

# Mutation-derived, genomic instability-associated lncRNAs are prognostic markers in gliomas

Shenglun Li*, Yujia Chen*, Yuduo Guo, Jiacheng Xu, Xiang Wang, Weihai Ning, Lixin Ma, Yanming Qu, Mingshan Zhang and Hongwei Zhang

Department of Neurosurgery, Sanbo Brain Hospital, Capital Medical University, Beijing, China
* These authors contributed equally to this work.

## ABSTRACT

**Background:** Gliomas are the most commonly-detected malignant tumors of the brain. They contain abundant long non-coding RNAs (lncRNAs), which are valuable cancer biomarkers. LncRNAs may be involved in genomic instability; however, their specific role and mechanism in gliomas remains unclear. LncRNAs that are related to genomic instability have not been reported in gliomas.

**Methods:** The transcriptome data from The Cancer Genome Atlas (TCGA) database were analyzed. The co-expression network of genomic instability-related lncRNAs and mRNA was established, and the model of genomic instability-related lncRNA was identified by univariate Cox regression and LASSO analyses. Based on the median risk score obtained in the training set, we divided the samples into high-risk and low-risk groups and proved the survival prediction ability of genomic instability-related lncRNA signatures. The results were verified in the external data set. Finally, a real-time quantitative polymerase chain reaction assay was performed to validate the signature.

**Results:** The signatures of 17 lncRNAs (*LINC01579, AL022344.1, AC025171.5, LINC01116, MIR155HG, AC131097.3, LINC00906, CYTOR, AC015540.1, SLC25A21. AS1, H19, AL133415.1, SNHG18, FOXD3.AS1, LINC02593, AL354919.2* and *CRNDE*) related to genomic instability were identified. In the internal data set and Gene Expression Omnibus (GEO) external data set, the low-risk group showed better survival than the high-risk group (*P* < 0.001). In addition, this feature was identified as an independent risk factor, showing its independent prognostic value with different clinical stratifications. The majority of patients in the low-risk group had isocitrate dehydrogenase 1 (*IDH1*) mutations. The expression levels of these lncRNAs were significantly higher in glioblastoma cell lines than in normal cells.

**Conclusions:** Our study shows that the signature of 17 lncRNAs related to genomic instability has prognostic value for gliomas and could provide a potential therapeutic method for glioblastoma.

Corresponding author
Hongwei Zhang,
zhanghongwei@ccmu.edu.cn

## INTRODUCTION

Genetic abnormalities and mutations are crucial factors in the development of cancer, resulting in a loss of balance in the nucleotide chain of most human tumors (*Hanahan & Weinberg, 2011*). Genomic instability is defined as an increased likelihood of acquiring chromosomal aberrations due to defects in processes such as DNA repair, DNA damage, replication, or chromosome segregation (*Tubbs & Nussenzweig, 2017*). Due to genomic disruption, genomic instability is typically subdivided into three categories: nucleotide, microsatellite, and chromosome (*Pikor et al., 2013*). The majority of tumor cells exhibit genomic instability. For example, more than two-thirds of human tumors gain or lose whole chromosomes during cell division. The abnormal expression of genes, such as *CDK4*, Ras downstream prokaryotic gene, and BRAF gene can contribute to this instability (*Pikor et al., 2013*; *Carvalho Claudia & Lupski James, 2016*). As a marker of cancer evolution, genomic instability (mainly caused by mutations in DNA repair genes) promotes cancer progression and has been identified as a key prognostic factor (*Kamata et al., 2010*; *Cui et al., 2010*). In hereditary cancers, genomic instability results from mutations in DNA repair genes and drives cancer development according to the mutator hypothesis. However, the molecular basis of genomic instability in sporadic (non-hereditary) cancers remains unclear. Recent high-throughput sequencing studies suggest that mutations in DNA repair genes are infrequent before therapy, arguing against the mutator hypothesis for these cancers (*Suzuki et al., 2003*). Additionally, in contrast to the different numbers of genetic mutations, various cancer types exhibit different patterns of somatic cell mutations, indicating specific carcinogenic mechanisms in tissues and cells (*Ottini et al., 2006*). It is, therefore, important to identify the potential molecular characteristics associated with genomic instability in cancer and explore their clinical significance.

Glioma, commonly known as glioma cerebri (GC), is a malignant primary brain tumor that accounts for approximately 81% of all tumors (*Negrini, Gorgoulis Vassilis & Halazonetis Thanos, 2010*). Glioblastoma (GBM), the most malignant among them, has an average survival time of just 16 months for patients (*Anandakrishnan et al., 2019*). GBM is characterized by uncontrolled cell proliferation, diffuse infiltration, necrosis, intense angiogenesis, strong resistance to apoptosis, and rampant gene instability (*Moini & Piran, 2020*). The etiology of glioma remains unclear; however, exposure to high doses of ionizing radiation, and genetic mutations associated with high penetrance of rare syndromes had been identified as two related risk factors in glioma tumorigenesis (*Stupp et al., 2005*). Thus, that it is critical to identify biomarkers associated with genomic instability to accurately evaluate the clinical prognosis of glioma patients.

Long non-coding RNAs are a type of RNA with over 200 nucleotides that are incapable of coding proteins (*Negrini, Gorgoulis Vassilis & Halazonetis Thanos, 2010*). Previous research has demonstrated that lncRNAs play a role in various life events (*Mattick John & Rinn John, 2015*; *Mercer Tim & Mattick John, 2013*) and biological processes that are thought to be involved in the development and progression of malignancies, including gliomas. These processes include stemness, angiogenesis, and drug resistance (*Shi et al.,*

*2017*). Their abnormal expression can affect cell proliferation, tumor progression, and metastasis (*Sanchez Calle et al., 2018*). In addition, abnormal lncRNA expression profiles in clinical glioma samples correlate with the degree of malignancy and histological differentiation, which has important clinical implications for the diagnosis and subclassification of gliomas (*Wang et al., 2016*; *Jing et al., 2016*). LncRNAs can act as molecular signaling mediators, regulating the expression of a specific set of genes and corresponding signaling pathways. For example, the *CRNDE-mTOR* pathway can promote the giloma growth (*Wang et al., 2015*) and lncRNAs have been found to play a role in various cancers. For example, *Ling et al. (2013)* reported a novel lncRNA, *CCAT2*, containing rs6983267 *SNP*. This lncRNA was highly overexpressed in microsatellite-stabilized colorectal cancer and could promote tumor growth, metastasis, or cause chromosome instability. This indicates that lncRNAs involved in the biological process of gene modification contribute to genomic instability and tumor progression. However, studies on the association of lncRNA with genomic instability are still lacking.

In this study, we identified a set of lncRNAs associated with genomic instability, and constructed a genomic instability-related lncRNA signature (GIrLncSig). We then validated its prognostic significance in glioma patients. The results showed that this signature had a great prediction role in the prognosis of glioma patients.

# MATERIALS AND METHODS

## Data collection

We collected the clinical characteristics, transcription group data, and somatic cell mutation data from GBM and low-grade glioma (LGG) patients. The datasets generated during this study are available in The Cancer Genome Atlas repository (https://portal.gdc.cancer.gov/). These data were matched using sample names. Patients who had no corresponding information about their survival, or who had less than 30 days of treatment were excluded to eliminate interference from non-cancer causes. We differentiated mRNA and lncRNAs using human genome profiles; mRNAs and lncRNAs were annotated using the HUGO Gene Nomenclature Committee (HGNC2) database (https://www.genenames.org/). A total of 629 samples retained paired lncRNA and mRNA expression profiles, and survival information, somatic mutation information, and common clinicopathological features were obtained for further study. All patients with glioma were randomly divided into two groups: training and test sets. A total of 316 patients in the training set were used to identify the prognostic features of lncRNAs and establish a risk model for the outcome. The test set included 313 patients and was used to independently validate the performance of the prognostic risk model. The GSE43378 dataset generated and analyzed during the current study is available in the Gene Expression Omnibus (GEO) repository (https://www.ncbi.nlm.nih.gov/geo/query/acc.cgi?acc=GSE43378) for the external validation of the model.

## Technical route

The study process is shown in Fig. 1. We collected and then analyzed the data from somatic cell mutations and transcription groups to obtain genomic instability-related lncRNAs

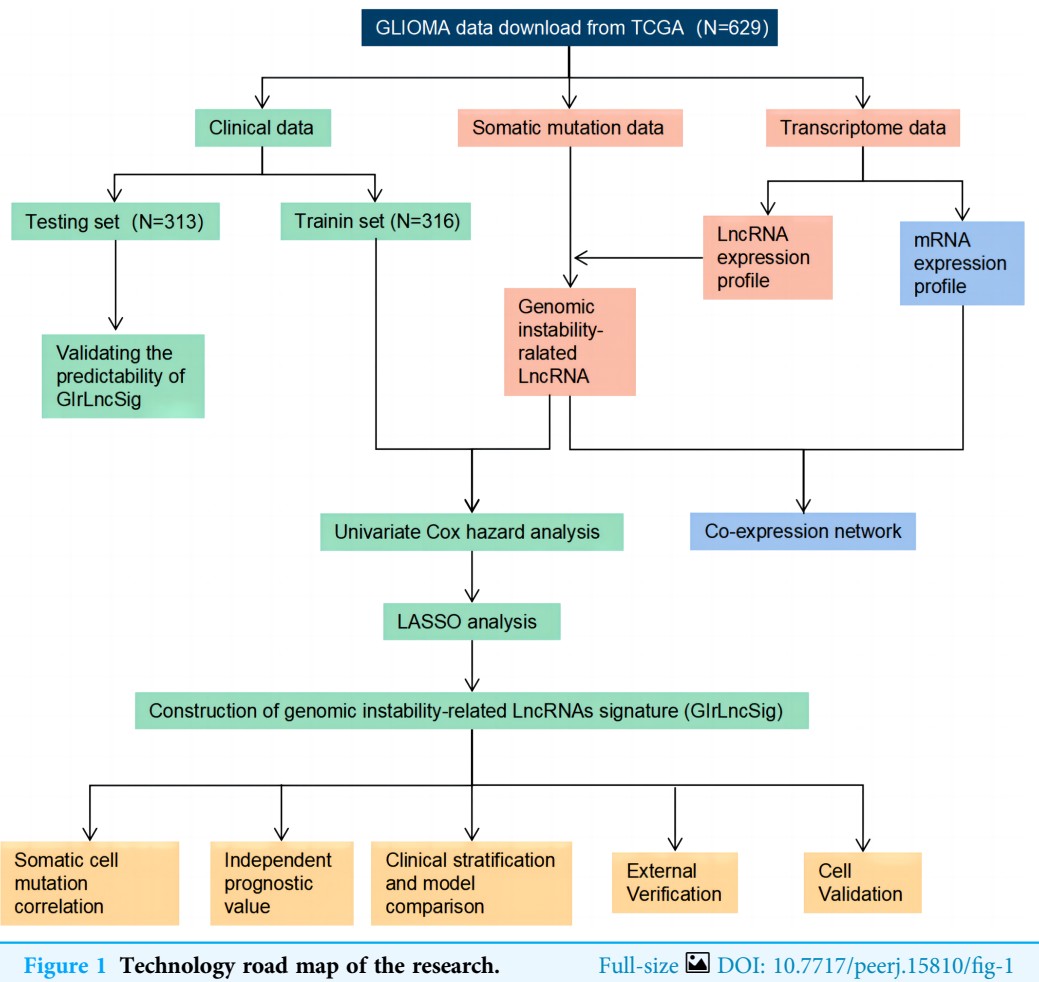

**Figure 1 Technology road map of the research.**

(GIrlncRNAs). The relationship between GIrlncRNAs and mRNA was then analyzed using co-expression analysis. Next, we randomly divided the patient cohort into training and testing sets. Cox and Lasso regression analyses of GIrlncRNAs were conducted to construct a prognostic signature. The signature was evaluated using mutation correlation analysis, model comparison, independent prognostic value analysis, clinical stratification, examination of the external dataset and cell line validation.

## Identification of GIrlncRNAs

We used a method from the Mutator Hypothesis (*Bao et al., 2020*) to identify GIrlncRNAs. Patients with the highest cumulative gene mutation count and who were at the lowest 25% cumulative gene mutation count were assigned to genomic unstable-like (GU) and genome stable-like (GS) groups, respectively. The average expression of lncRNAs between the two groups was compared using the Wilcoxon rank-sum test in the limma package of the R software. The cut-off thresholds were intended to be |fold change| > 2.0 and false discovery rate (FDR) < 0.05.

## Construction of GIrLncSig

The "survival" software package of R was used to conduct univariate Cox regression analysis on the training set to assess the relationship between the expression level of GIrlncRNAs and the overall survival time of patients. The least absolute shrinkage and selection operator (LASSO) regression algorithm was used to further screen candidate GIrlncRNAs to construct the GIrlncRNAs prognostic signature (GIrLncSig). The following formula, based on a combination of the Cox coefficient and gene expression, was used to calculate the signature risk score:

$$GIrLncSig\ score = \sum_{i=1}^{n} coefi \times Ei$$

GIrLncSig is the prognostic risk score for patients with glioma. Ei represents the expression level of lncRNAi in patients and coefi represents the coefficient of lncRNAi. The median GIrLncSig score was used as the risk cut-off point to divide glioma patients into low-risk and high-risk groups. The survival curves of the two groups were plotted by Kaplan-Meier method using "Survminer" and "Survival" packages in R language, and a log-rank sum test that obtained $P < 0.05$ was considered significant.

## Real time-PCR validation

The U87, U251, LN229, and U343 cell lines of human glioblastoma and the immortalized cell line SVGp12 were used for the cellular level validation of lncRNA in the model. All cells were cultured in DMEM supplemented with 10% FBS, 100 U/ml penicillin, and 100 U/ml streptomycin. The cultures were maintained at 37 °C in a humidified environment containing 5% carbon dioxide and were confirmed to be mycoplasma-free prior to experimental use.

A total of 1 ml Trizol (Invitrogen, Waltham, MA, USA) was added to the collected cells to extract total cellular RNA, and the absorbance value of RNA at 260 nm was measured using a Nanodrop 2000 UV spectrophotometer. Then, 1 ug of RNA was removed and used to synthesize cDNA by reverse transcription (New England Biolabs, Ipswich, MA, USA), the SYBR Green (Applied Biosystems, Foster City, CA, USA) method and CFX96 real-time PCR system (Bio-Rad, Hercules, CA, USA) were used for real-time polymerase chain reaction (RT-PCR), and actin was used as an internal control. Amplification was performed at 95 °C/120 s followed by 39 cycles at 95 °C/5 s and 60 °C/30 s. Relative expression of RNA was calculated using the $2^{-\Delta\Delta Ct}$ method. The primers were generated by Sangon Biotechnology (Shanghai, China). The primers for lncRNAs in GIrLncSig are listed in Table 1.

# RESULTS

## Identified genomic instability-related lncRNAs (GIrlncRNAs) in glioma samples

To identify genomic instability-related lncRNAs, we sorted glioma patients with the number of gene somatic mutation sites and defined the top 25% ($n = 227$) as the genome unstable (GU) group and the last 25% ($n = 110$) as the genome stable (GS) group. Next, we
**Table 1  qPCR primers designed to amplify mRNA of lncRNAs in GIrlncSig as risk factor.**

| LncRNA | Forward | Reverse |
| --- | --- | --- |
| LINC01579 | 5′-TCCCAGTGAAGAGAGAGCGA-3′ | 5′-CTAAGTTCCACGTCACGGCT-3′ |
| LINC01116 | 5′-GAATGGCAAAGCACTTGGGG-3′ | 5′-AGCTCTCCTTGCAGGTAGGT-3′ |
| MIR155HG | 5′-AGGGGTTTTTGCCTCCAACT-3′ | 5′-TCTTTGTCATCCTCCCACGG-3′ |
| CYTOR | 5′-TTCCAACCTCCGTCTGCATC-3′ | 5′-AATGGGAAACCGACCAGACC-3′ |
| H19 | 5′-GACATCTGGAGTCTGGCAGG-3′ | 5′-CTGCCACGTCCTGTAACCAA-3′ |
| SNHG18 | 5′-TGCACTTTGCCACTGCTACA-3′ | 5′-GGGGAATGTGGTTCTCCCTT-3′ |
| FOXD3.AS1 | 5′-AAGAGTAAGAGCAGCGCACC-3′ | 5′-ACCTGAGTGGTTTGGTTGGG-3′ |
| CRNDE | 5′-ATTCAGCCGTTGGTCTTTGA-3′ | 5′-CTTCTGCGTGACAACTGAGGA-3′ |

compared the expression profiles of lncRNAs in these two groups to identify the lncRNAs that were significantly different. After screening, we identified 91 differentially expressed lncRNAs (Table S1) and selected the top forty significantly different lncRNAs between the GS and GU groups for heatmap plotting (Fig. 2A). These 91 differentially expressed lncRNAs were significantly associated with genomic instability; thus, they were defined as genome instability-related lncRNAs (GIrlncRNAs). Consensus cluster analysis was then performed on 896 samples from The Cancer Genome Atlas (TCGA) collection (including 390 GBMs and 506 LGGs), and all samples were divided into two groups based on the differential expression of GIrlncRNAs and the median number of accumulated somatic mutations (Fig. 2B). The group with the higher number of accumulated somatic mutations was defined as the GU-like group and the group with a lower number of accumulated somatic mutations was defined as the GS-like group. The two clustered groups had significantly different somatic mutation patterns (Fig. 2C). We analyzed the expression correlation between lncRNAs and their target mRNAs. We also selected the top nine strongly correlated mRNAs as target genes since lncRNAs cannot perform direct biological functions but can regulate mRNAs. We then constructed a co-expression network from our results (Fig. 2D).

### Construction of GIrlncRNAs signature in the training set

A total of 629 glioma patients from the TCGA project were randomly divided into a training dataset ($n = 316$) and a test dataset ($n = 313$). Patients' clinicopathological characteristics are shown in Table 2. Eighty prognostic related lncRNAs were identified based on univariate Cox proportional risk regression in the training set (Table S2 and Fig. S2). Lasso regression analysis and stepwise multifactor Cox proportional risk regression analysis were performed since lncRNAs have distinct biological functions and many lncRNAs interact with each other. These steps were taken because there were large amounts of data and the potential for inaccurate model construction. A total of 17 lncRNAs were screened as independent prognostic factors to build the prognostic model (Figs. 3A, 3B and Table 3). A prediction model was finally obtained: genomic instability-related lncRNA signature (GIrLncSig) score = (0.0441217761138621 × *LINC01579*) +(−0.197814767909076×*AL022344.1*) + (0.0387926268692573 ×

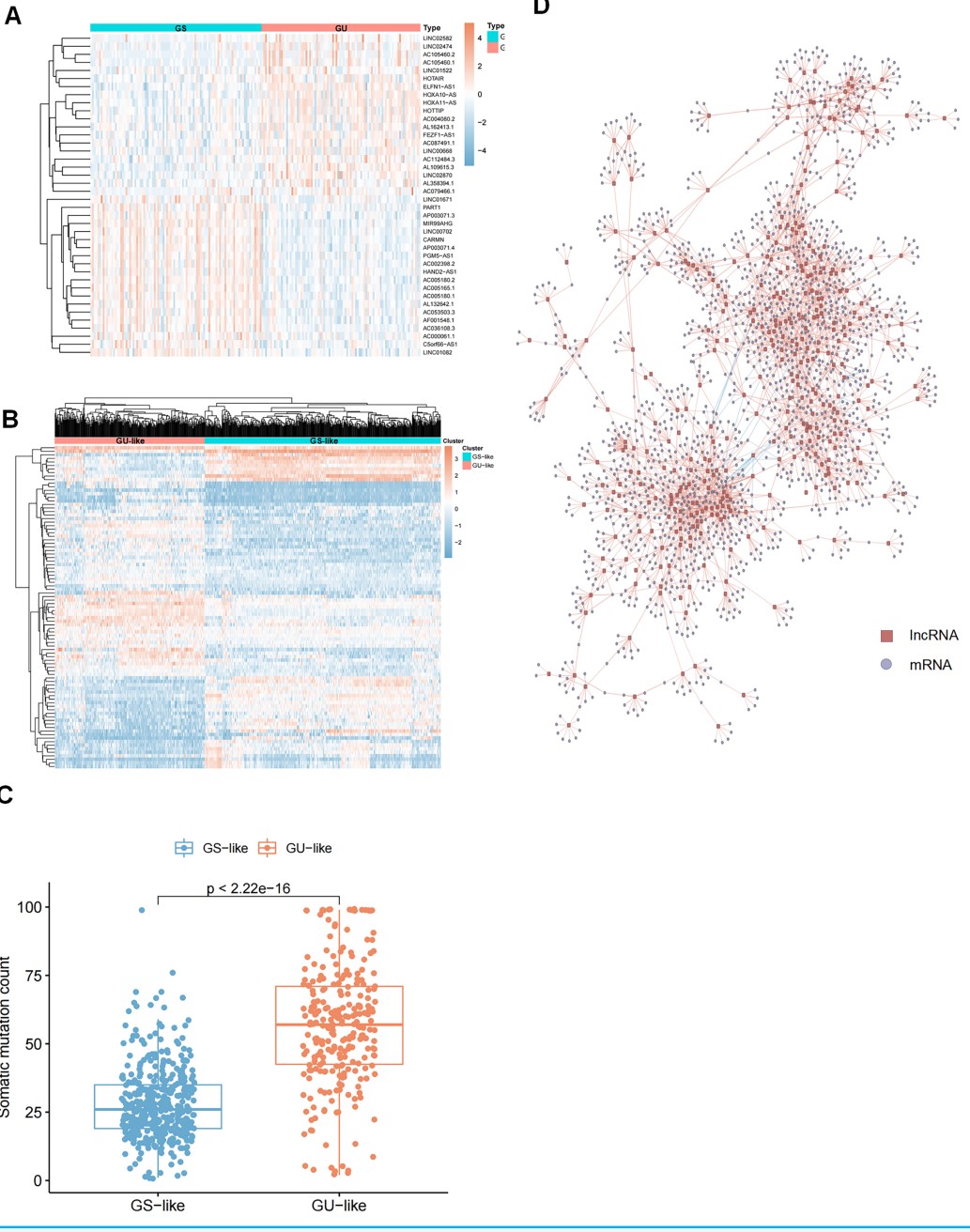

**Figure 2 Selection of lncRNAs associated with genomic instability in GC patients and demonstration of their target genes.** (A) Heat map of the expression of 40 of the most significantly different (20 each of up- and down-regulated expression) lncRNAs in GU and GS groups. (B) Unsupervised clustering of 896 GC patients based on 91 GIrlncRNAs expression patterns. The red cluster on the left is a GS-like cluster, and the blue one on the right is a GU-like cluster. (C) Comparative box plots of cumulative mutation counts in somatic cells. The number of mutations in GU-like group was significantly higher than that in the GS-like group ($P < 0.001$, Mann-Whitney U test). (D) Co-expression network of lncRNAs and mRNAs associated with genomic instability based on Pearson correlation coefficients. Red circles represent lncRNAs, and blue circles represent mRNAs.

**Table 2 Clinicopathological features of glioma patients in each set.**

| Covariates | Type | Total ($n = 629$) | Test ($n = 313$) | Train ($n = 316$) | P value |
|---|---|---|---|---|---|
| Age | <=65 | 547 (86.96%) | 276 (88.18%) | 271 (85.76%) | 0.4338 |
| | >65 | 82 (13.04%) | 37 (11.82%) | 45 (14.24%) | |
| Gender | Female | 270 (42.93%) | 136 (43.45%) | 134 (42.41%) | 0.8538 |
| | Male | 359 (57.07%) | 177 (56.55%) | 182 (57.59%) | |
| Tumor grade | G2 | 189 (30.05%) | 101 (32.27%) | 88 (27.85%) | 0.5808 |
| | G3 | 176 (27.98%) | 88 (28.12%) | 88 (27.85%) | |
| | G4 | 264 (41.97%) | 124 (39.62%) | 140 (44.3%) | |

Note:
Chi-squared test, $P < 0.05$ means significantly different.

*AC025171.5) + (0.000726291473736049 × LINC01116) + (0.140438109893274 × MIR155HG) + (0.205877895933553 × AC131097.3) + (−0.0369769918481854 × LINC00906) + (0.0588040388194378 × CYTOR) + (−0.0336307964944069 × AC015540.1) + (−0.108512558356276 × SLC25A21.AS1) + (0.00613294332460668 × H19) + (0.0819381463964954 × AL133415.1) + (0.00495065493579931 × SNHG18) + (0.0180933234213193 × FOXD3.AS1) + (−0.0263113038864061 × LINC02593) + (0.0103411968758757 × AL354919.2) + (0.0460814562382004 × CRNDE).* A positive coefficient for lncRNAs suggested that high expressions were associated with long survival times as a protective factor in glioma. In contrast, a negative coefficient for lncRNAs indicates that it is a risk factor for glioma. The risk score of each patient in the training set was obtained using GIrLncSig, and then these patients were divided into high- and low-risk groups based on the median risk score (0.0103411968758757). Survival analysis revealed that the low-risk group had significantly better survival rates than the high-risk group (Fig. 3C). The 5-year survival rates were 4.43% in the high-risk group and 14.56% in the low-risk group. The ROC curve showed that the AUC was 0.934, 0.898, and 0.904 at 1, 3, and 5 years, respectively (Fig. 3D). We sorted patients in the training set according to their risk score and there were different expression levels of GIrlncRNAs in the two groups (Fig. 3E). Patients with high-risk scores exhibited an increased expression of risk genes and decreased expression of protective genes, whereas those with low-risk scores exhibited the opposite trend. We also found a significant difference in somatic mutation patterns between patients in the high-risk and low-risk groups, implying that the model could accurately reflect the somatic mutation situation in gliomas (Fig. 3F).

## Independent validation of GIrLncSig in the glioma cerebri data set of transcription data and external validation in the GEO data set

An independent TCGA test set of 313 patients was used to examine the performance of GIrLncSig. The test set used the same GIrLncSig and risk thresholds as the training set and the 313 patients in the test set were divided into high-risk ($n = 173$) and low-risk groups ($n = 140$). There was a significant difference in overall survival (Fig. 4A). The overall survival rate was significantly lower in the high-risk group than in the low-risk group, which is consistent with the results of the training group. The 5-year survival rate was

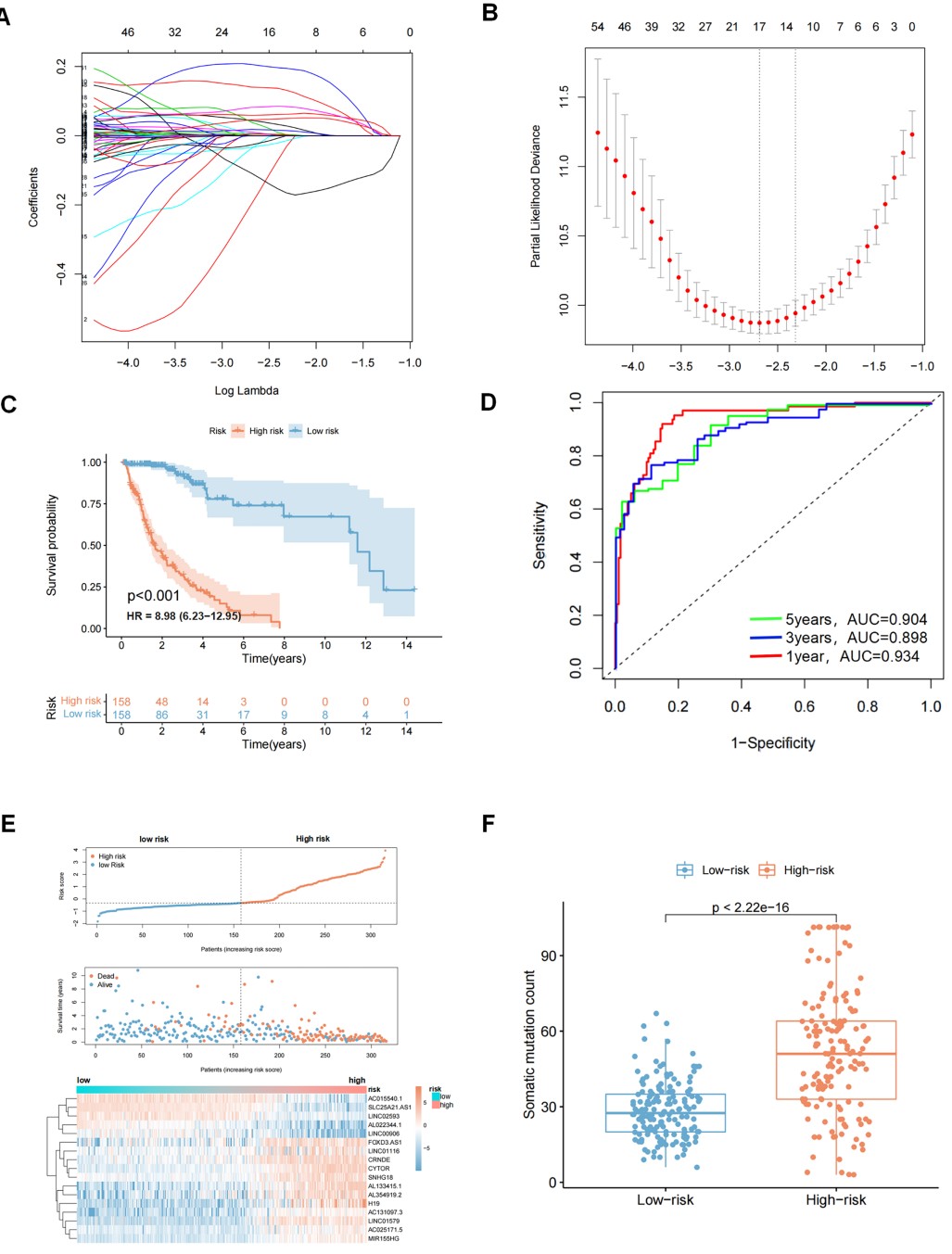

**Figure 3 LASSO analysis and to evaluate and validate the predictive performance of genomic instability-related lncRNA signature (GIrLncSig) on the overall survival of GC patients in the training set.** (A) Distribution of lasso coefficients is plotted. When Log Lambda equals −2.7, 17 variables are retained. (B) Distribution of partial likelihood deviation of lasso coefficients. Seventeen variables were retained when bias likelihood deviation was minimized (Log Lambda = −2.7). (C) Log-rank test, $P < 0.05$. (D) ROC curves of GIrLncSig for predicting 1, 3, and 5-year survival in the training set. (E) A set of risk maps, including risk score maps, survival distribution maps, and lncRNAs expression heatmaps, were used for the training set. As GIrLncSig score increased, the expression of lncRNAs and patient death rate also changed. (F) Box plots comparing somatic mutation counts between high and low-risk groups in the training set (Mann-Whitney U test, $P < 0.01$).

**Table 3 The 17 prognositic-related GIrlncRNAsobtained by LASSO analysis.**

| Gene | Description | Coef |
|---|---|---|
| LINC01579 | Long intergenic non-protein coding RNA 1579 | 0.0441 |
| AL022344.1 | *Homo sapiens* chromosome 10 clone XX-Y214H10, *** SEQUENCING IN PROGRESS ***, 3 unordered pieces | −0.1978 |
| AC025171.5 | *Homo sapiens* chromosome 5 clone CTD-2035E11, WORKING DRAFT SEQUENCE, 16 unordered pieces | 0.0388 |
| LINC01116 | Long intergenic non-protein coding RNA 1116 | 0.0007 |
| MIR155HG | MIR155 host gene | 0.1404 |
| AC131097.3 | *Homo sapiens* BAC clone RP11-789L24 from 2, complete sequence | 0.2059 |
| LINC00906 | Long intergenic non-protein coding RNA 906 | −0.0370 |
| CYTOR | Cytoskeleton regulator RNA (C2orf59; LINC00152) | 0.0588 |
| AC015540.1 | *Homo sapiens* clone RP11-385G16, *** SEQUENCING IN PROGRESS ***, 89 unordered pieces | −0.0336 |
| SLC25A21.AS1 | SLC25A21 antisense RNA 1 | −0.1085 |
| H19 | H19 imprinted maternally expressed transcript | 0.0061 |
| AL133415.1 | *Homo sapiens* chromosome 10 clone RP11-124N14, *** SEQUENCING IN PROGRESS ***. | 0.0819 |
| SNHG18 | Small nucleolar RNA host gene 18 | 0.0050 |
| FOXD3.AS1 | FOXD3 antisense RNA 1 | 0.0181 |
| LINC02593 | Long intergenic non-protein coding RNA 2593 | −0.0263 |
| AL354919.2 | Long intergenic non-protein coding RNA AL354919.2 | 0.0103 |
| CRNDE | Colorectal neoplasia differentially expressed gene | 0.0461 |

7.51% in the high-risk group, which is lower than the 16.4% in the low-risk group (Fig. 4A). ROC curve analysis showed that the AUC of GIrLncSig was 0.875 for 1 year and 0.773 for 3 years (Fig. 4B). The expression levels of GIrLncSig, patient death distribution counts and model lncRNA expression are shown in Fig. 4C. A significant difference was observed in the somatic mutation pattern between patients in the high-risk and low-risk groups (Fig. 4D), and this result was similar in the training group. To further validate the prognostic significance of GIrLncSig, a cross-platform was performed in other independent datasets from different platforms. GSE43378, a dataset from the Gene Expression Matrix data set, was downloaded for further analysis because of the large sample size and complete clinicopathological features. We investigated the relationship between glioma and genomic instability in this independent dataset and found that four lncRNAs (Linc01116, CRNDE, Linc00906, and SNHG18) in GIrLncSig were included in GSE43378. The expressions of Linc01116 and CRNDE were positively correlated with tumor grade (Figs. 4E and 4F) and the survival time was significantly different between their high- and low-expression subgroups (Figs. 4G and 4H). These results were consistent with those observed in the training and test sets.

## Evaluation of independent prognostic significance of GIrLncSig and clinical stratification analysis

In the TCGA dataset, the prognosis of GIrLncSig was analyzed by adjusting for clinical stratification, including age (>65 and ≤65 years), gender (male and female), tumor classification (WHO grade II, III and IV), and other clinical factors. In all clinical

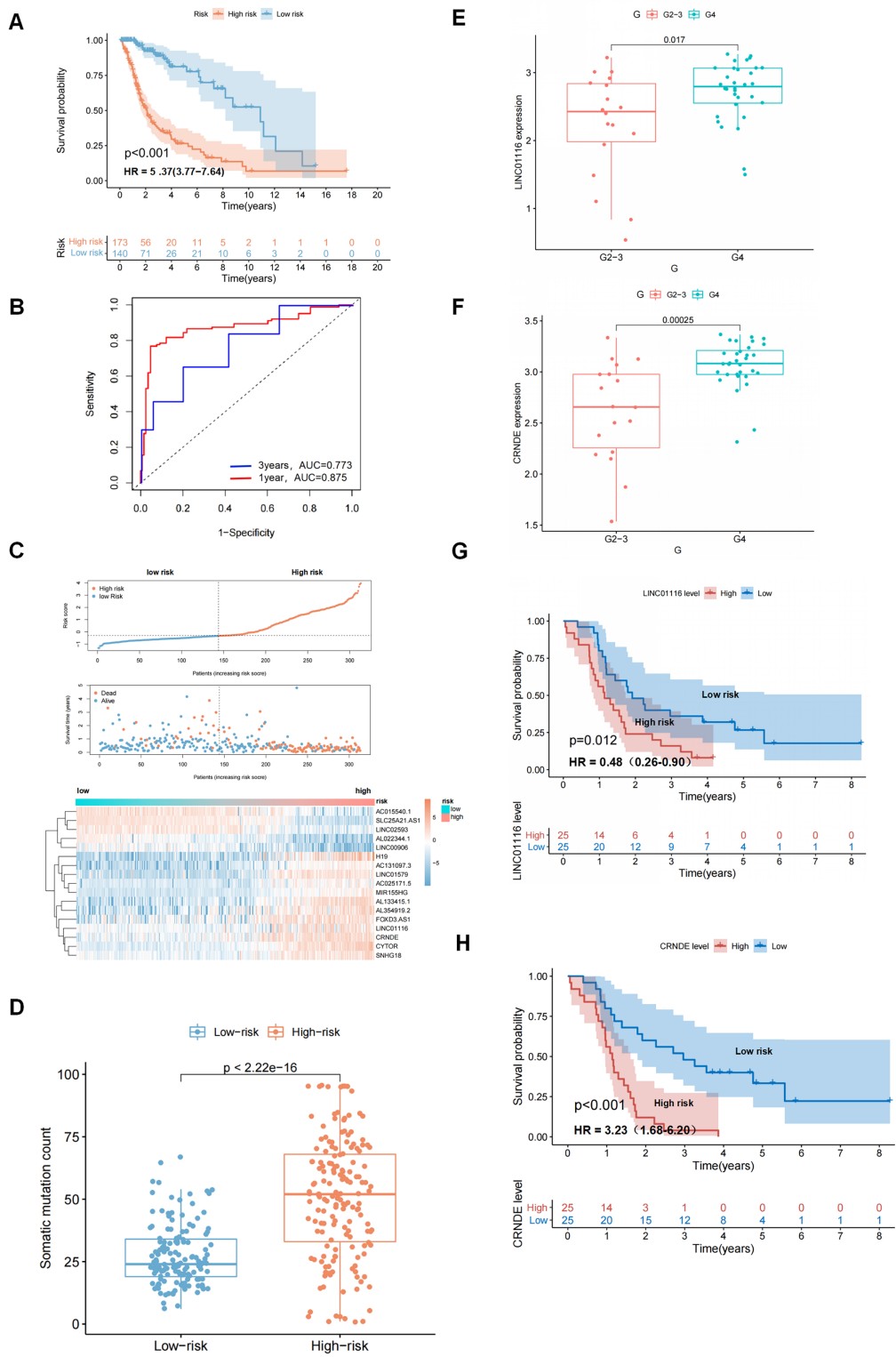

**Figure 4** **Performance evaluation of GIrLncSig in TCGA set and testing set of GC patients and GEO dataset was subjected to external validation.** Kaplan-Meier survival curves for patients in high-risk and low-risk groups classified by GIrLncSig score in testing set. (A) Patients in the low-risk group had prolonged survival compared with the high-risk group (log-rank test, *P* < 0.05). ROC curves of GIrLncSig predicting 1-year and 3-year survival in the testing set (B). Regarding risk score plots, survival distribution plots, and lncRNAs expression heatmaps for the testing set (C), the expression of lncRNAs and

**Figure 4** (continued)
patient mortality changed with increasing GIrLncSig scores. The box plot compares the number of somatic mutations in high-risk and low-risk groups in the testing set (D) (Mann-Whitney U test, $P < 0.01$). LINC01116 expression in high-grade and low-grade gliomas in GEO dataset (E) and expression of CRNDE (F). Survival analysis was performed according to grouping of high expression of LINC01116 (G) and CRNDE (H), and there was a difference in survival between the two groups (log-rank test, $P = 0.012$, $P < 0.001$).               

subgroups, the survival rate in the low-risk group was higher than that in the high-risk group (Fig. 5). This demonstrated that GIrLncSig exhibited a significant independent prognostic prediction value on the overall survival of glioma patients under different clinical stratification conditions.

## GIrLncSig was associated with IDH1 mutation status and comparison between GIrLncSig prediction and other lncRNA model predictions

We observed a difference in the *IDH1* status in a cohort of glioma patients. *IDH1* is a security mutant in the nervous region, as reported, therefore we determined that it was important to evaluate the connection between GIrLncSig and *IDH1* mutation status. We compared the differences between the high-risk and low-risk groups on the training set and test set and the results revealed that the proportion of *IDH1* mutation was significantly higher in the low-risk group than in the high-risk group, regardless of whether the patient was affected by glioma cerebri (GC) (Fig. 6A) or low-grade glioma (Fig. 6C). These results imply that patients with *IDH1* mutations were at lower risk. We then divided the patients into four subgroups (*IDH1* mutation/GS-like, *IDH1* mutation/GU-like, *IDH1* wild-type/GS-like and *IDH1* wild-type/GU-like) to consider both the GIrLncSig and *IDH1* mutation status. As shown in Figs. 6B and 6D, there were significant differences in the survival time among the four groups. This suggests that *IDH1* mutation status might be related to genomic instability and combining them together is better for predicting prognosis in glioma patients.

We then compared the predictability of GIrLncSig and two recently reported lncRNA signatures for survival prediction using the same TCGA glioma patients. Compared to the 10-lncRNA prognostic signature reported by *Pan et al. (2020)* (PanlncSig), 4-lncRNA prediction signature reported by *Li et al. (2019)* (LiDlncSig), 5-lncRNA prediction signature reported by *Li et al. (2021)* (LiXlncSig), 10-lncRNA prediction signature reported by *Luan et al. (2019)* (LuanlncSig), and 9-lncRNA prediction signature reported by *Tao et al. (2021)* (TaolncSig) as displayed in Fig. 6E, our GIrLncSig had an AUC value of 0.889 in 1-year OS. These results were more effective than those of PanLncSig (AUC value = 0.852), LiDlncSig (AUC value = 0.835), LiXlncSig (AUC value = 0.790), LuanlncSig (AUC value = 0.842), and TaolncSig (AUC value = 0.862) in predicting patient survival. In summary, the above results confirmed the reliability and effectiveness of GIrLncSig in predicting GC patients.

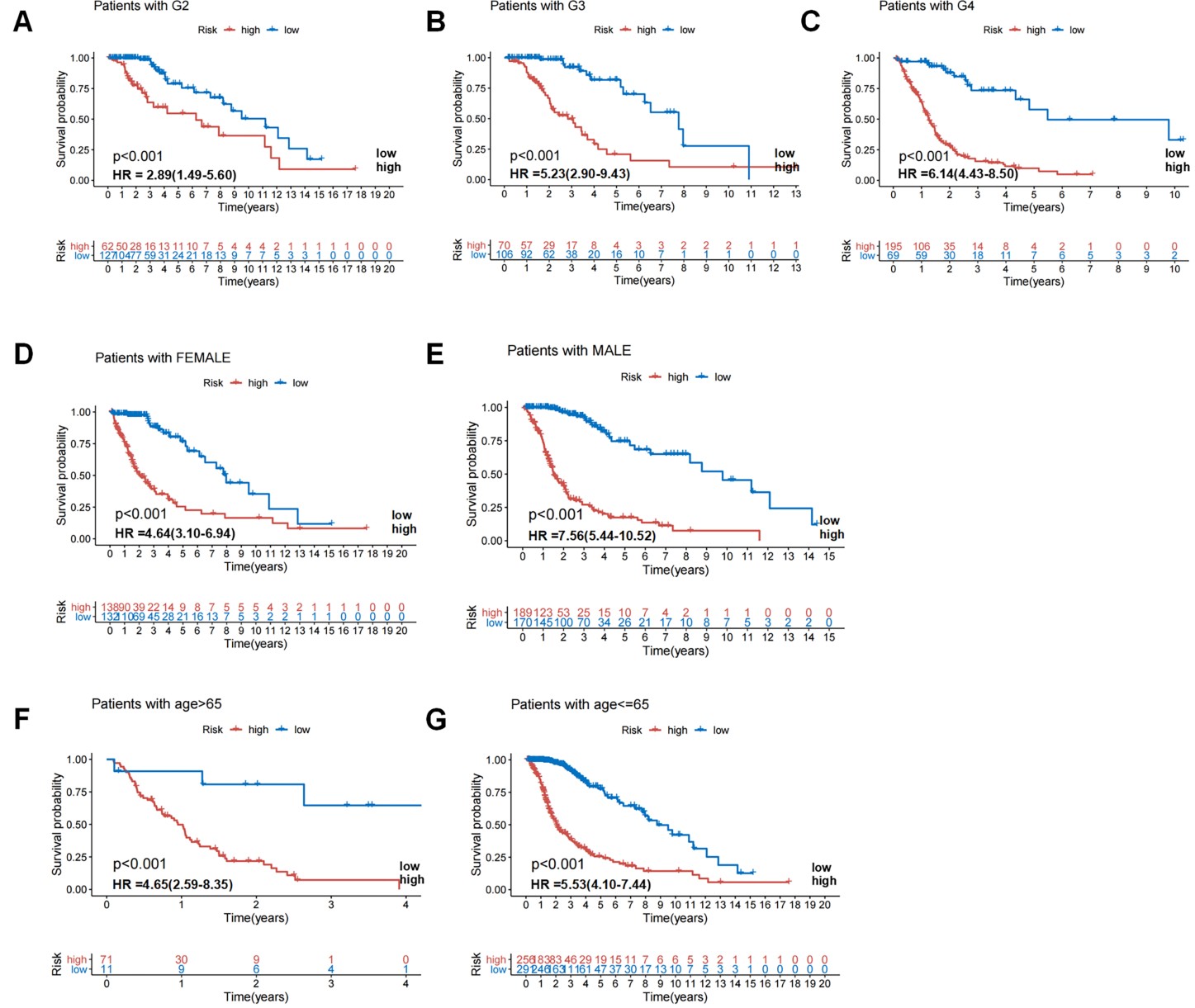

**Figure 5** **Patients in high-risk and low-risk groups of GC were clinically stratified by age, gender, and tumor grade for survival differences.** Kaplan-Meier survival curves of patients in high-risk and low-risk groups were analyzed for seven clinically stratified subgroups, including tumor grade II (A), tumor grade III (B), tumor grade IV (C), female (D), male (E), high-risk group (F), and low-risk group (G). In all clinically stratified subgroups, patients in low-risk group had better survival outcomes than those in high-risk group (log-ranch test, $P < 0.001$).

## Validation of GirLncSig

To further validate the prognostic significance of GirLncSig in gliomas, we verified eight lncRNAs in GirLncSig in four glioblastoma cell lines (U87, U251, LN229, and U343) and one normal cell line (SVGp12). The expression levels of these eight lncRNAs were significantly higher in glioblastoma cell lines than in normal cells (Fig. 7A). Similarly, the results of clinical samples also proved that the expression of GirLncSig in normal brain

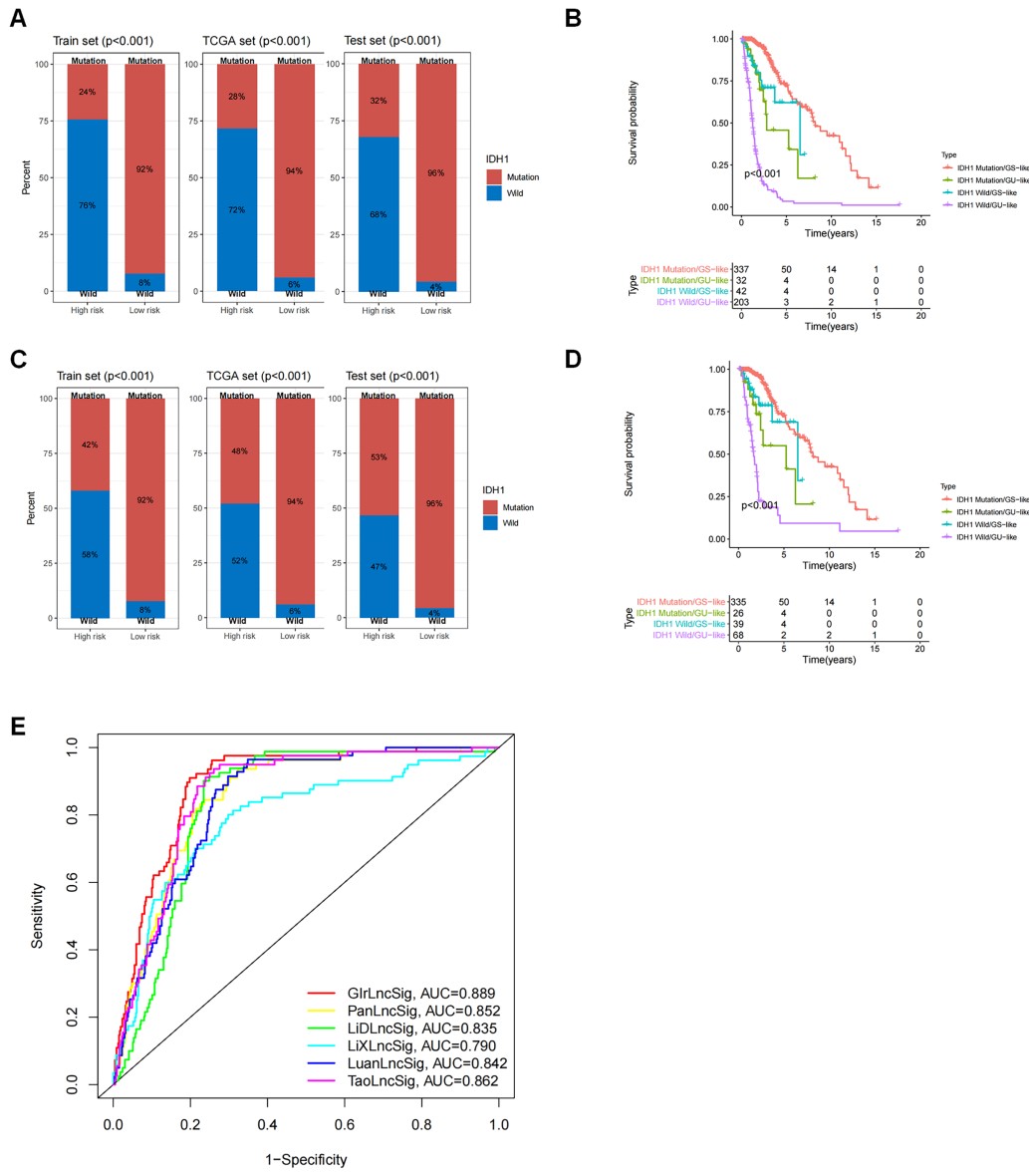

**Figure 6 Correlation analysis of GInLncSig and IDH1 mutation status.** Box plots of the proportion of IDH1 mutations in all glioma patients (A) and patients with low-grade gliomas (C) in high-risk and low-risk groups (Chi-square test, $P < 0.05$). Kaplan-Meier survival curves for IDH1 mutation and GIrLncRNAs grouping of all glioma patients (B) and low-grade glioma patients (D) revealed statistically significant differences in overall survival between the four groups (log-ranking test, $P < 0.001$). (E) ROC curves for 1-year survival prediction for GIrLncSig and two other existing signatures.

tissue (N1–N3) was lower than that in glioma tumor tissue (T1–T6), further validating the prognostic significance of GirLncSig.

We then examined the relationship between GirLncSig and cell proliferation and found that in the glioma samples from the TCGA database, the expression of almost all model-related sub-risk lncRNAs was positively correlated with the expression of cell proliferation markers *Ki67* and *PCNA* (Figs. 8A and 8B). Similarly, we also detected *CDH2*

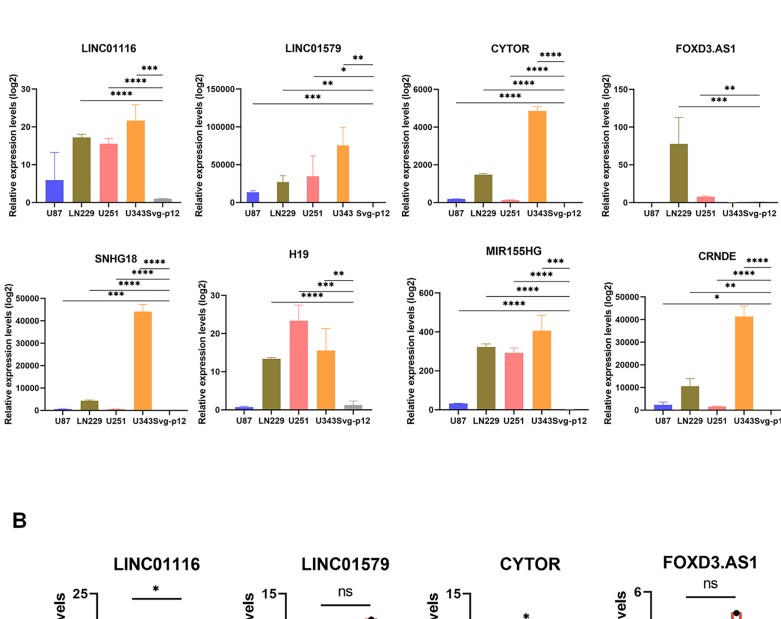

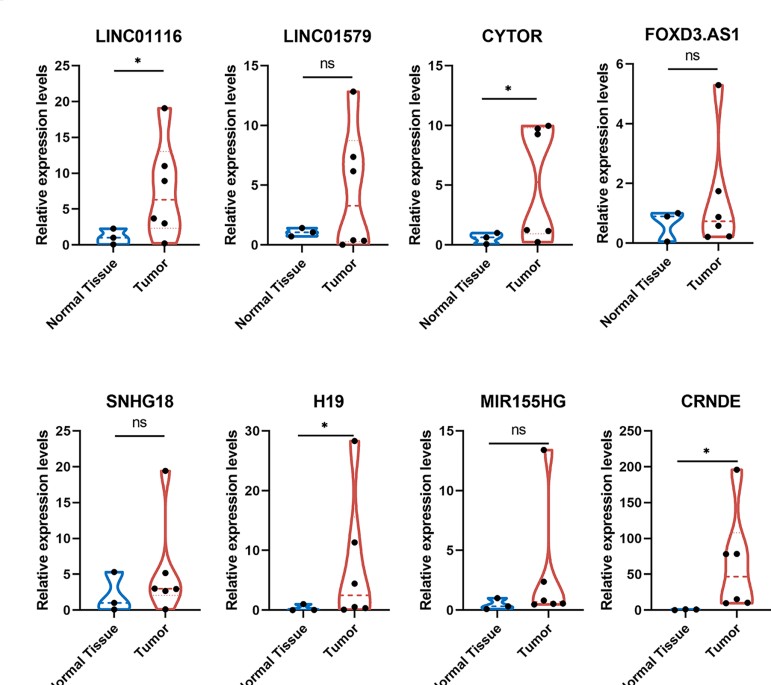

**Figure 7 RT-PCR validation of cell lines and clinical samples.** (A) The expression of eight risk factors lncRNAs in cellular validation differed between tumor and normal cells (unpaired T-test, $P < 0.05$). (B) The expression of eight risk factors lncRNAs in clinical samples (Mann Whitney test). $^*P < 0.05$, $^{**}P < 0.01$, $^{***}P < 0.001$, $^{****}P < 0.0001$; ns, not significant.

and *VIM*, which are markers of cell migration, invasion, and mesenchymal transition. The eight lncRNAs were found to be positively correlated (Figs. 8C and 8D). These findings tentatively show that GIrLncSig can promote tumors by affecting cell proliferation and EMT.

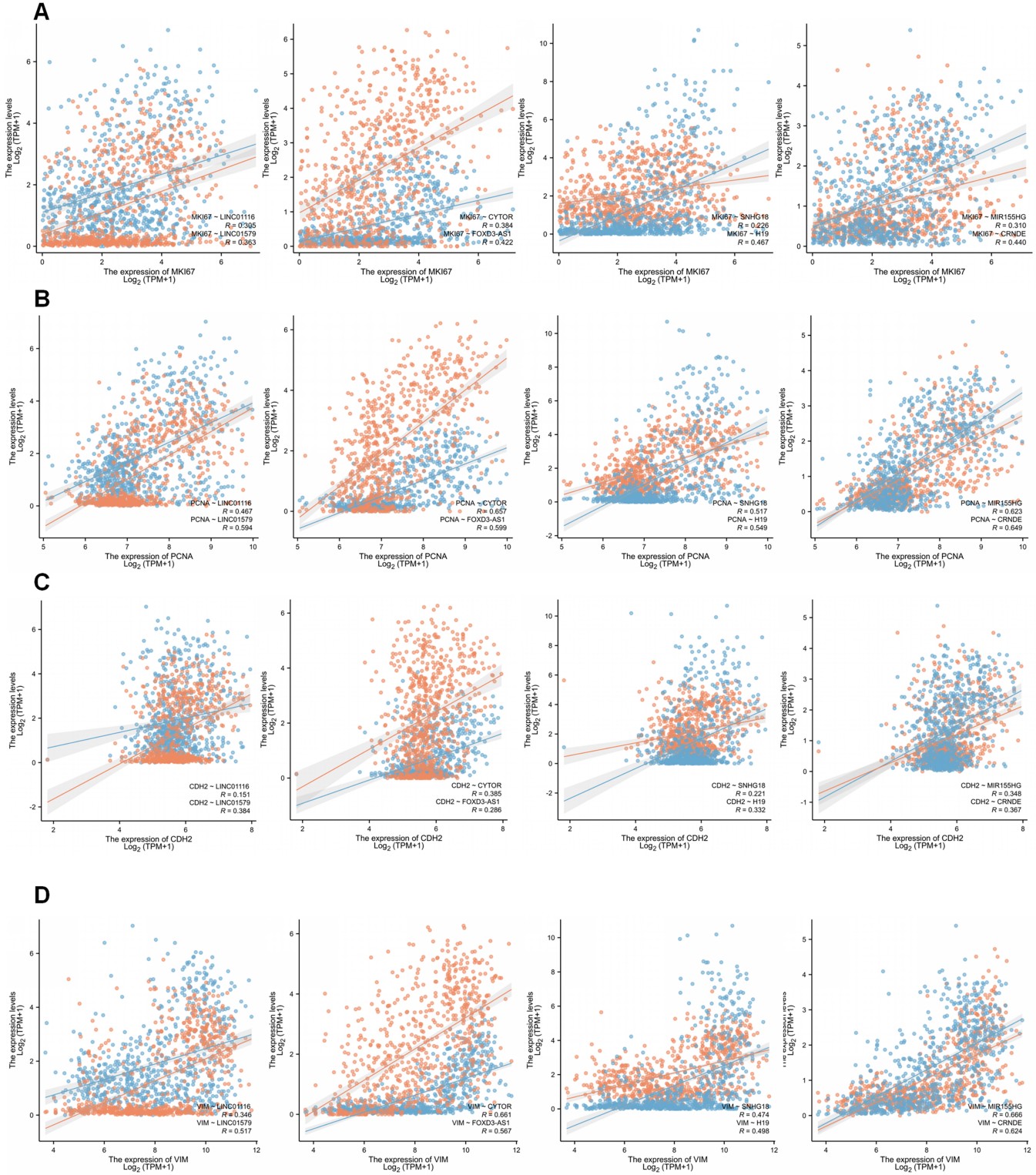

**Figure 8 Correlation of eight risk factor lncRNAs and cell phenotypes of patients with glioma in the TCGA database.** (A) The correlation between eight risk factor lncRNAs and cell proliferation marker mki67. (B) Correlation with cell proliferation marker PCNA. (C) Correlation with the mesenchymal transformation gene CDH2 and (D) correlation with the mesenchymal transformation gene Vimentin (Pearson, *P* < 0.001).

## DISCUSSION

Genomic instability is caused by chromosomal segregation errors during mitosis that result in aneuploidy mutations of whole chromosomes in daughter cells, or by DNA damage that causes chromosome structure changes. These changes can result in gene translocations, deletions, inversions, and breaks (*Carvalho Claudia & Lupski James, 2016*; *Wickramasinghe & Venkitaraman Ashok, 2016*). Genomic changes can occur at different levels, from mutations in a single or few nucleotides to the gain or loss of entire chromosomes. This may trigger abnormal divisions, multinucleation, and tripolar mitosis. Maintaining genetic integrity is critical for cell viability and is accomplished through complex repair processes. When these processes are defective, genomic instability occurs, resulting in the accumulation of chromosomal mutations which can then cause susceptibility to cancer (*Mackay et al., 2018*; *Rajendran & Deng, 2017*). Genomic instability plays a fundamental role in cancer progression and recurrence, suggesting that its pattern and degree have significant diagnostic and prognostic implications (*Zhang et al., 2019*; *Rancoule et al., 2017*). LncRNAs have recently been demonstrated to be promising tumor biomarkers. Abnormal expression of lncRNAs in tumors is associated with disease progression and may serve as a prognostic marker for patients (*Kronenwett et al., 2006*; *Mettu et al., 2010*; *Gupta et al., 2010*). Furthermore, recent advances in understanding the functional mechanisms underlying lncRNAs have recognized that lncRNAs are essential for genomic stability (*Huarte & Rinn, 2010*; *Prensner et al., 2011*).

Glioma is the most common primary intracranial tumor. To date, no studies have examined the lncRNA signature of genomic instability in glioma. In this study, we identified a group of GIrlncRNAs in GC and determined their significance for predicting patient survival. We then identified 91 GIrlncRNAs and compared their expression levels in different mutation counts. Systematic clustering analysis and subsequent differential analysis of mutation counts confirmed the association between these lncRNAs with genomic instability. Based on co-expression with 91 GIrlncRNAs, we further investigated whether GIrlncRNAs could predict the clinical outcomes of glioma patients by constructing a GIrLncSig consisting of 17 GIrlncRNAs (*LINC01579, AL022344.1, AC025171.5. LINC01116, MIR155HG, AC131097.3, CYTOR, AC015540.1, SLC25A21.AS1, H19, AL133415.1, SNHG18, FOXD3.AS1, LINC02593, AL354919.2,* and *CRNDE*). Using GIrLncSig we classified patients into two risk groups with a significant difference in survival in the training set, which was validated on an independent test set. Similar results were found in the external GEO dataset, GSE43378. These validation results across multiple datasets and technology platforms indicate that GIrLncSig may be an indicator of genomic instability in cancer patients. Few studies have been conducted on tumor models based on genomic instability lncRNA. Recent studies were selected for glioma lncRNA models, including those by *Li et al. (2019)* and *Pan et al. (2020)*, to construct lncRNA models using the lncRNA expression data of patients in the CGGA database and GSE16011 dataset, respectively. None of these studies discussed the impact of genomic instability on tumors, nor did they use the TCGA database. In addition, *Li et al. (2021)* studied the prognostic model of immune related lncRNA. *Tao et al. (2021)* explored the

relationship between epithelial mesenchymal transition (EMT)-related lncRNA and the prognosis of glioma. Similarly, *Luan et al. (2019)* studied the autophagy-related lncRNAs. The prognostic model of lncRNAs was also mentioned in *Kiran et al. (2019)*'s article, but it was for low-grade glioma and did not cover GBM samples. In addition, *Zheng et al. (2021)*'s article studied the role of barbed wire-related lncRNAs in glioma. However, no study has linked lncRNAs to genomic instability. This study investigates genome instability, since genome instability and ferroptosis affect cancer at different magnitudes. Following the first discovery of glioblastoma (GBM) in 2008 by the Johns Hopkins University team in the United States using whole exome sequencing technology, *IDH1-R132* was detected in 18 cases (12%) of 149 tumor samples. The *IDH1* mutations detected in the study are more common in young patients with secondary tumors, and the median overall survival (OS) of patients with mutations reaches 3.8 years, which is much better than the median OS of 1.1 years in IDH1 wild-type patients (HR = 3.7, $P < 0.001$) (*Munschauer et al., 2018*; *Hu et al., 2018*). In our study, the frequency of IDH1 mutations was significantly higher in the low-risk group than in the high-risk group, reflecting the reliability of the model.

This study creatively constructed a prediction model related to genomic instability through the analysis of patients' genomic mutation data, transcriptome data, and clinical data in the TCGA database. This model has an obvious leading role in predicting the prognosis of glioma patients (Fig. 6E). It may play a role in the survival difference of glioma patients in high- and low-risk groups (Figs. 4A–4C) and in different grades of glioma (Figs. 5A–5C). The value of this model in the prognosis and survival of glioma patients has been demonstrated. In addition, we validated the expression levels of eight GIrlncRNAs with known sequences in four glioma cell lines, and the results revealed significant differences in the expression levels of eight lncRNAs as risk factors between glioma cell lines and ordinary cells.

Although our study showed some links between genomic instability and prognosis in patients with glioma, several limitations remain. We lacked the validation of the function of our GIrLncSig to ensure its accuracy and reproducibility. Therefore, further studies are necessary and its action mechanism in glioma development and progression remains to be further investigated.

## CONCLUSION

In summary, we constructed a risk prediction signature consisting of 17 lncRNAs associated with genomic instability. This signature can predict the prognosis of glioma patients and reveal their genomic instability. It serves as a reference indicator for clinical individualized treatment and has great significance for glioma patients. Therefore, this signature may be an important tool for further investigating the role of lncRNAs in genomic instability. Further research will clarify the relationship between lncRNAs and genomic instability in tumors.

### Funding
The authors received no funding for this work.

### Competing Interests
The authors declare that they have no competing interests.

### Author Contributions
- Shenglun Li conceived and designed the experiments, performed the experiments, analyzed the data, prepared figures and/or tables, authored or reviewed drafts of the article, and approved the final draft.
- Yujia Chen conceived and designed the experiments, performed the experiments, analyzed the data, prepared figures and/or tables, authored or reviewed drafts of the article, and approved the final draft.
- Yuduo Guo conceived and designed the experiments, analyzed the data, prepared figures and/or tables, authored or reviewed drafts of the article, and approved the final draft.
- Jiacheng Xu conceived and designed the experiments, analyzed the data, prepared figures and/or tables, and approved the final draft.
- Xiang Wang conceived and designed the experiments, analyzed the data, prepared figures and/or tables, and approved the final draft.
- Weihai Ning analyzed the data, prepared figures and/or tables, and approved the final draft.
- Lixin Ma conceived and designed the experiments, authored or reviewed drafts of the article, and approved the final draft.
- Yanming Qu conceived and designed the experiments, prepared figures and/or tables, authored or reviewed drafts of the article, and approved the final draft.
- Mingshan Zhang conceived and designed the experiments, prepared figures and/or tables, authored or reviewed drafts of the article, and approved the final draft.
- Hongwei Zhang conceived and designed the experiments, authored or reviewed drafts of the article, and approved the final draft.

### Data Availability
The raw data is available in the Supplemental Table.

### Supplemental Information
Supplemental information for this article can be found online at http://dx.doi.org/10.7717/peerj.15810#supplemental-information.

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
