# Peer review of "Mutation-derived, genomic instability-associated lncRNAs are prognostic markers in gliomas"

_PeerJ, doi:10.7717/peerj.15810_

## Round 0.1 · original submission · Major Revisions

Dear Dr. Li,

Thank you for submitting your manuscript "Mutator-derived lncRNAs drive genomic instability and represent a kind of prognostic marker for glioma" to PeerJ. We have now received reports from the reviewers, and, after careful consideration internally, we have decided to invite a major revision of the manuscript.

As you will see from the reports copied below, the reviewers raise important concerns. We find that these concerns limit the strength of the study, and therefore we ask you to address them with additional work. Without substantial revisions, we will be unlikely to send the paper back for review.

If you feel that you are able to comprehensively address the reviewers’ concerns, please provide a point-by-point response to these comments along with your revision. Please show all changes in the manuscript text file with track changes or color highlighting. If you are unable to address specific reviewer requests or find any points invalid, please explain why in the point-by-point response.

Thanks

Abhishek Tyagi, PhD
Academic Editor,
PeerJ

Reviewer 1 ·

Basic reporting

line 73 has spelling mistake (constracted)
line154
Several spelling mistakes were identified and need to be corrected
Grammar need to be checked.

Experimental design

Need to improve

Validity of the findings

need improvement

Additional comments

Manuscript Title "Mutator-derived lncRNAs drive genomic instability and represent a kind of prognostic marker for glioma"

Title of the manuscript doesn't provide a strong information about this work please avoid "a kind of" in the title Mutator-derived lncRNAs drive genomic instability and represent a kind of prognostic marker for glioma"

what kind of IDH1 mutations are more frequent? Is there any reported hotspot mutations were identified?

Line 100 "Next, we randomly divided patient cohort into training and testing sets." At what basis samples were divided? It is important to classify them in specific manner.

Instead of using "genome unsteady (GU)/ steady group" the authors consider to use Genomic unstable/stable group.

This line is not carry clear meaning " Next, we compared the expression profiles of lncRNAs in these two groups to identify the lncRNAs that were significantly different".
what kind of significant difference between two groups?

When integrating GBM 390 and LGG 506 two cohorts, mutational landscape is significantly difference between two cohort, then how they used two group?

A prediction model is little confusing. The authors need to provide clear formula instead of example data.


Line274 In addition to reference 27, It is relevant to cite the following paper (doi: 10.18632/oncotarget.17225) discussed mutational drivers in cancer "When the repair link is defective,genomic instability occurs, resulting in the accumulation of chromosomal mutations which can cause cancer susceptibility"

Discussion is lack of information about overall outcomes of this classification towards GBM prognosis
Similarly conclusion is not sound. "For the clinical hierarchical management and individualized treatment, it has great significance of glioma patients."

Almost all figures are not in readable size and good resolution. The authors need to provide clear figures.
Minor



Table 3 need Gene Description and coefficient need to be just 4 decimal point is enough.

Reviewer 2 ·

Basic reporting

Li et al. identified lncRNAs associated with the mutation burdens in glioblastoma and glioma samples, and used the expression of top lncRNAs to predict the prognosis of patients. I find the motivation of this study needs more support and a key question needs to be answered before the results can be correctly interpreted. See below for details.

Experimental design

1. A primary goal of the study is to predict the prognosis of glioma/glioblastoma. Instead of searching the whole transcriptome for marker genes, the authors focused on a subset, namely the lncRNAs. While lncRNAs are related to genome instability, I am sure that many protein coding genes can be tied to genome instability as well. Therefore I find the motivation of this study is relatively weak. More analyses are required to show that the lncRNAs, rather than the whole transcriptome, are more likely to contain good markers for predicting prognosis and genome instability, and/or to inform the underlying mechanisms.

2. I am concerned with decision of pooling GBM and LGG samples during analysis. The fact that TCGA treats them as two types of cancer suggests that they can be quite different. Have the authors checked the numbers of mutations and prognosis of GBM vs LGG? The differentially expressed lncRNAs between the GS and GU samples could reflect a difference between GBM and LGG samples rather than a difference in genome stability. Before this possibility is excluded, it is difficult to interpret the results of this study.

Validity of the findings

The title claims “mutator-derived lncRNA drive genomic instability”, but the study does not address any causal relation between the lncRNA and genomic instability.

Additional comments

Several texts are confusing:
1. line 34. Not sure what the internal dataset refers to, because all datasets seem to be retrieved from previous publications.
2. line 72. “genomic instability of lncRNA” is weird.
3. line 88. Did HUGO provide the clinicopathological features, survival information, somatic mutations taken?
4. Line 150. Why do the top 25% include 227 samples but the bottom 25% include only 110 samples?
5. Line 157 - 160. What is the difference between the 896 glioma samples and the 629 samples mentioned at line 85? How was the expression of lncRNAs used to classify the samples? Based on line 160-161, the authors used the number of mutations to classify samples and merely plotted the expression of lncRNAs in the two groups.

Reviewer 3 ·

Basic reporting

The focus of this manuscript is to identify genomic instability-related lncRNAs and explore their prognostic value in gliomas. The manuscript is overall well-written. I only have a few suggestions for improvement.

1. The prognostic value of lncRNAs in gliomas has previously been reported (e.g., Kiran, Manjari, et al. "A prognostic signature for lower grade gliomas based on expression of long non-coding RNAs.", PMID: 30392137, Zheng, Jianglin, et al. "A prognostic ferroptosis-related lncRNAs signature associated with immune landscape and radiotherapy response in glioma.", PMID: 34095147). The manuscript should illustrate the limitations of the previous studies and the major differences between the present study and the previous works.

2. The Introduction part is inadequate. The prognostic value of LncRNAs in gliomas and other types of cancers should be better introduced.

Experimental design

no comment

Validity of the findings

1. To validate the prognostic significance of GIrLncSig in gliomas, the authors verified eight
lncRNAs in GIrLncSig in four glioblastoma cell lines. Why were only the expression level of 8 of the 17 lncRNAs investigated? To further validate the prognostic significance of the IncRNAs, does their expressions correlate with the cell proliferation, invasion, or migration of the glioblastoma cell lines?

2. Figure quality needs be greatly improved. The text on some of the figures are too blurry to read.

Reviewer 4 ·

Basic reporting

1. The introduction needs more detail. For example, genome instability generally refers to xxxx, or genetic abnormalities include xxx types, including xx, xx, and xx.

What is the relationship between genome instability and mutation? (the description at lines 45: "genetic abnormalities and mutations are crucial factors in cancer"; the description at lines 50: genomic instability (mainly caused by mutations in DNA repair genes))

2. Many figures are not high quality, well labelled or described, and they are difficult to be read. For example, Figure 2A,2B,2D; Figure 3E, Figure 4C, Figure 5,Figure 6A,6C. The font size, font color and figure resolution should be revised.

3. The number of sample from the TCGA dataset is not consistent in the paper. For example, the description "896 samples from the cancer genome atlas (TCGA) collection (including GBM 390 and LGG 506)" at line 157 (page 11), however "a total of 629 glioma patients from TCGA project were randomly divided into a training dataset 170 (n=316) and a test dataset (n=313)" at line 169 (page 11) and Figure 1. The relationship between 629 and 896 should be described.

4. The writing should be improved. For example, the blank space between two words are missing ("withlong" at line 188, "timeas" at line 188, "0.904at" at 194, "survivaltime" at 222, "trainingt" at 224, ). Some words are not consistent ("seventeen lncRNAs" at line 40 and figure 3 legend, "17 lncRNAs" at many other lines; "Fig 1" at line 97 and other figures starts with "Figure"). In addition, some punctuation symbols are not right ("." at line 41, "," before (https) at line 82). And the gene names should be italicized.

Experimental design

no comment

Validity of the findings

5. LGG and GBM samples often exhibit distinct histological, clinical (e.g. survival time) and genomic features (e.g. mutation count). The proportion of LGG and GBM samples in each group defined by the authors should be described clear.

---

## Round 0.2 · Minor Revisions

Dear Dr. Li,

Your manuscript still requires a number of Minor Revisions.

Abhishek Tyagi
PeerJ

Reviewer 1 ·

Basic reporting

All suggested comments were addressed by the authors.

Change Cancer biomarker in Keyword section. It is mentioned as "biomaker"

Several minor errors are still exists. Grammar, spelling errors, space between two words etc. (few examples line 42, 196, 419)

Experimental design

All suggested comments were addressed by the authors.

Validity of the findings

All suggested comments were addressed by the authors.

Additional comments

Make sure the manuscript is free from grammar and spelling errors. I found several mistakes throughout the manuscript.

Reviewer 3 ·

Basic reporting

see additional comments

Experimental design

see additional comments

Validity of the findings

see additional comments

Additional comments

The authors have conducted additional investigation and revised the manuscript to address all my concerns.

---

## Round 0.3 · Minor Revisions

Dear Dr. Li,

Thank you for your submission to PeerJ.

The Section Editor has commented and said:

"The manuscript still needs significant editing for English as well as clarity.

In addition, it is not clear how the authors evaluate “genomic instability associated lncRNAs”. All that I find in a section called “Technical route” is: "The process of this study is shown in Figure 1. After collecting the data, we analyzed data from somatic cell mutations and transcription groups to obtain genomic instability-related lncRNAs (GIrlncRNAs)." How does this work?

Also, there was a section "Identification of GIrlncRNAs" To identify GIrlncRNAs, a method derived from the Mutator Hypothesis was applied. Patients with the highest cumulative mutation count and the lowest 25% were assigned to genomic unstable-like (GU) and genome stable-like (GS) groups respectively." Do you mean, mutation of genes or lncRNAs? "

Please address these concerns.


With kind regards,
Abhishek Tyagi
Academic Editor
PeerJ Life & Environment

---

## Round 0.4 · Minor Revisions

Dear Dr. Li,

This manuscript is scientifically sound but still needs editing for English and is not acceptable in its current form.

With kind regards,
Abhishek Tyagi
Academic Editor
PeerJ Life & Environment

---

## Round 0.5 · accepted · Accept

Dear Dr. Li,

Thank you for your submission to PeerJ. I am pleased to Accept it for publicaiton
Abhishek Tyagi